# Microfluidic Microwave Sensor Loaded with Star-Slotted Patch for Edible Oil Quality Inspection

**DOI:** 10.3390/s22176410

**Published:** 2022-08-25

**Authors:** Xueyun Han, Yingping Zhou, Xiaosong Li, Zhongjun Ma, Lei Qiao, Chenghao Fu, Peidong Peng

**Affiliations:** 1College of Electronics and Electrical Engineering, Henan Normal University, Xinxiang 453000, China; 2School of Environment, Henan Normal University, Xinxiang 453000, China

**Keywords:** microwave, sensor, dielectric permittivity, resonant frequency

## Abstract

In this paper, we present a new microfluidic microwave sensor loaded with a star-slotted patch for detecting the quality of edible oil. The relative dielectric permittivity and the quality of edible oil will change after being heated at a high temperature. Therefore, the quality of edible oil can be detected by measuring the relative dielectric permittivity of edible oil. The sensor is used to determine the edible oil with different dielectric permittivity by measuring the resonance frequency offset of the input reflection coefficient, which operates at 2.68 GHz. This sensor is designed based on a resonant approach to provide the best sensing accuracy and is implemented using a substrate integrated waveguide structure combined with a pentagonal slot antenna operating at 2.3~2.9 GHz. It can detect greasy liquids with the real part of the complex permittivity ranging from two to three.

## 1. Introduction

Deep-frying is a common cooking method that has been used since ancient times, and the color, aroma, crispness, and taste of deep-fried foods are very popular among people. When frying food, the temperature of the oil is usually heated to 170 °C or even higher, at which point the cooking oil will produce many chemical reactions that will cause many toxic volatile and non-volatile compounds in the fried food. People in modern society are fonder of fried food than they ever were in the past. The quality of fried food mainly depends on the nature and type of oil, but the quality of frying oil is uneven, and most of them are used after repeated heating. After the oil is heated, it will be consumed and degraded, leading to a series of chemical reactions [1,2,3]. These repeatedly heated oils can cause chronic diseases and serious damage to people’s health [1,2,3,4,5]. Therefore, it is necessary to test the quality of cooking oil.

Recently, a number of studies on the quality testing of oils were undertaken. Most scholars have studied the effects of heating temperature, heating duration, and heating frequency on the quality of edible oils [4,5]. These studies are mainly through the analysis of acid value, oxidation value and other content to determine the quality of edible oil [2,3,4,5]. However, this method is not very economical and needs to be analyzed by professional technicians. The use of solvents also causes environmental pollution. S. Rubalya Valantina et al. found that the determination of dielectric permittivity of edible oils can better obtain the characterization and reusability of oils [4,5,6]. In recent years, with the rapid development of microwave technology, many scholars have started to consider the use of microwave technology to detect the quality of various substances. Microstrip line technology is a non-contact technology that can remotely detect and analyze liquids with different dielectric permittivity by detecting changes in resonant frequencies [7,8,9,10], therefore, microwave technology is a topic of interest for various scholars. Thomas Chretiennot et al. designed a sensor specifically for monitoring the glucose concentration in water by using microwave and microfluidic technology, and it was identified as a promising technique for monitoring blood parameters [11].

In this paper, a microstrip patch microwave sensor with a size of 70 mm × 70 mm is designed, which covers the frequency band from 2.3 GHz to 2.9 GHz. When the edible oil is injected into the microfluidic channel, the sensor can detect the edible oil with different permittivity by the offset of resonant frequency to determine the quality of edible oil. The microstrip patch microfluidic sensor, which is simple and easy to design and manufacture, has a good application prospect and market value in the quality inspection of edible oil.

## 2. Structural Design

The performance of the sensor is largely affected by the material and width of the dielectric substrate, therefore, the material and thickness of the dielectric substrate need to be determined in the design process. The relative permittivity and losses tangent of the material are two electrical characteristic parameters to be prioritized. If the dielectric constant is too large, the size of the patch will be reduced, and the bandwidth of the patch will also be reduced. If the dielectric constant is too small, the edge field around the patch will increase, thus reducing the radiation efficiency of the sensor. Meanwhile, the stability of the dielectric constant of the substrate is also very important, a change in the dielectric constant can cause a shift in the frequency of the patch. A dielectric substrate with high loss will reduce the efficiency of the antenna and increase the feedback loss. After a comprehensive analysis, epoxy resin (FR4_epoxy) and polydimethylsiloxane (PDMS) with relative permittivity of 4.4 and 2.8, and loss tangent of 0.02 and 0.002, respectively, are used as dielectric substrates in this design. FR4 is a common material in antenna design with good electrical properties, and its performance is little affected by the environment. PDMS is a flexible material with high dielectric properties. The microfluidic channel is formed by etching in PDMS for pumping the liquid to be measured. The resonant frequency of the sensor will change when a liquid with different dielectric constants flows in the microfluidic channel. A thick dielectric substrate not only improves the mechanical strength of the sensor, increases the radiated power, and reduces the conductor loss but also broadens the frequency band. However, the thick substrate will increase the dielectric loss and cause significant excitation of surface waves. The thickness of the substrate should follow [12]
(1)$$h\leq \frac{c}{4f\sqrt{\varepsilon -1}}$$
where *c* is the speed of light; *f* is the operating frequency; and *ε* is the dielectric constant of the substrate. Usually at *h*/*f* < 0.1, it is ensured that no significant excitation of surface waves is caused.

The structure of the microfluidic sensor designed in this paper is shown in Figure 1, which is composed of three layers. Figure 1a–c show the slotted circular radiation patch on the top layer, the dielectric substrate in the middle, and the copper-plated ground on the bottom, respectively. Figure 1d shows the structure of the PDMS microfluidic channel. The middle dielectric substrate is also a three-layer structure, the upper and lower layers are made of 70 × 70 × 0.3 mm epoxy resin (FR4_epoxy) substrate, and the middle is a 0.8 mm thick polydimethylsiloxane (PDMS) substrate. A microfluidic channel of cross-section size 3 × 0.4 mm is etched inside the PDMS substrate [13,14,15,16]. The radiation patch designed in this paper is a circular slotted antenna with a radius of 30 mm [17].

The shape and size of the radiation patch and the material of the dielectric substrate directly affect the performance of the sensor. This is mainly because the edge field from the radiation patch to the dielectric substrate will cause electromagnetic radiation from the patch to the substrate. In this design, the feed mode of the microstrip line is used to provide a signal source for the patch [18,19,20]. This feeding mode is used because the feeder is coplanar with the microstrip antenna patch, which is easy to produce. The feeder also has radiation, which can reduce the gain of the antenna. The shape and size of the microstrip patch, the width, length, and position of the slot can affect the antenna input impedance, change the resonant frequency of the sensor, and thus affect the performance of the sensor. The position of the slot affects the electric field distribution of the sensor, which makes the sensor more sensitive to the change of dielectric permittivity, thus improving the sensitivity of the sensor. 

The optimization steps of the shape of the sensor patch are shown in Figure 2. Starting from the conventional circular patch antenna designed for the center frequency, the optimization of the slot of the patch and the optimization of the slot of the microstrip line are carried out successively. The circular patch was chosen because it has a single parameter for generating frequency, the radius of the patch. The effective radius of the circular antenna is calculated by Equation (2) [21].
(2)$$R_{e}=R{\left\{1+\frac{2H}{\pi \varepsilon_{r}R}\left[ln\left(\frac{\pi R}{2H}\right)+1.7726\right]\right\}}^{1/2}$$

In order to improve the sensitivity of the sensor, the liquid to be measured needs to be placed under a strong electric field. Slotting the sensor can increase the electric field strength at the slotted area. In this paper, we choose to slot a pentagram because the slotted area is irregular, which can make the sensor obtain a strong electric field. The slotting can also reduce the return loss to improve the sensor’s information transmission performance. The S11 parameters corresponding to each optimization step are shown in Figure 3. After slotting, the center frequency of the sensor shifts to the lower frequencies, and the amplitude of the corresponding return loss decreases for each step. The smaller the return loss, the better the measurement performance of the sensor becomes. Figure 4 shows the electric field distribution corresponding to each step of the designed sensor. It can be seen that the electric field of the slotted sensor is stronger compared to the circular sensor, and the strong electric field is mainly concentrated at the microstrip line, the slotted edge and the edge of the circular patch. These concentrated electric field distribution places can maximize the electric field disturbance, and the microstrip channel position of this design is below the maximum electric field (confirmed from the color code). The slotting on both sides of the microstrip line is to obtain a better return loss for the sensor and improve its measurement performance.

Compared with the traditional patch antenna, this sensor has a pentagram-shaped slot in the middle of the radiating patch and another slot next to the feed line, so that the antenna has a good electric field strength and a narrow bandwidth. The size of the slotting has a large impact on the performance of the sensor. In this paper, the side length of the pentagram *L*_1_ and the side slot length of the feeder *L*_2_ are optimized, and the best size is finally selected. As shown in Figure 5, the resonant frequency of the sensor gradually decreases with the gradual increase in the pentagram side length *L*_1_, but the effect on the reflection coefficient amplitude is small. Figure 6 shows the variation of the return loss with the length of *L*_2_ at the center frequency point of the sensor. For the antenna, the sensor has a good reflected signal when its return loss is less than −10 dB, so the optimal length of *L*_1_ is 11.7 mm and the optimal length of *L*_2_ is 9 mm as can be seen from Figure 5 and Figure 6. After the optimization of the above steps, the structural parameters of the sensor are shown in Table 1.

## 3. Simulation

ANSYS High-Frequency Structure Simulator (HFSS) is applied to simulate the sensor in this paper. When a liquid with a different dielectric constant is pumped into the microchannel, the working frequency of the sensor will change differently. When resonance occurs, the sensor establishes a strong electric field. The liquid in the microchannel flowing over this region can change the distribution of the local electric field and thus affect the resonant behavior of the sensor. Specifically, the resonant frequency and bandwidth are influenced by the complex dielectric constant of the liquid sample [22,23]. Note that the resonant frequency is also partly affected by the exact location of the microfluidic channel. However, this is not a problem because the measurements are relative, and the position of the channel does not change. When the quality of edible oil changes, its permittivity will also change, but not very distinct. 

To investigate whether the sensor is feasible for detecting the quality of edible oil, we change the permittivity of microchannels and use HFSS to simulate microchannels with different permittivity to test the sensitivity of the sensor. The permittivity of the microfluidic channel was increased from two to three and by 0.2 units each time. The simulation results are shown in Figure 7. From Figure 7, the operating frequency of the sensor decreases gradually with the increase in the dielectric constant, in a linear relationship. As the dielectric constant of the microfluidic channel gradually increases, the charge polarization within the dielectric medium causes an increase in the capacitance of the sensor, which can lead to a shift in the operating frequency of the sensor toward lower frequencies. However, when the permittivity of the microfluidic channel is 2.8, the shift of resonant frequency decreases because the permittivity of the microfluidic channel is equal to the permittivity of the PDMS, therefore affecting the offset of the operating frequency of the sensor. In general, the permittivity of most edible oils changes between 2.4 and 3 before and after heating [24,25]. Therefore, only the dielectric constants of two to three were simulated in this paper. To investigate the effect of the change of dielectric loss factor of oil on the reflection coefficient of the sensor, the dielectric constant of the microfluidic channel is kept constant in the simulation, and the size of the dielectric loss tangent angle of the microfluidic channel is gradually increased from 0 to 0.03, and the dielectric constant is increased by 0.01 each time. The relationship between the S11 parameters of the sensor and the variation of dielectric loss tangent angle of the microchannel is shown in Figure 8. The results show that the dielectric loss has no effect on the resonant frequency of the sensor but has a certain effect on the amplitude of the reflection coefficient, and the amplitude of the reflection coefficient increases with the increase in the loss factor, but the change is not very obvious. The edible oils chosen as examples in this paper are rapeseed oil and soybean oil which are most commonly used. When rapeseed oil and soybean oil are not heated, the dielectric permittivities are 2.41 and 2.64, respectively. However, their dielectric permittivities rise to 2.61 and 2.83, respectively, after being heated at 170 °C for 12 h. The heating of edible oil not only changes its dielectric constant but also has a greater impact on its loss factor. As the heating temperature increases, the dielectric constant of the oil increases faster, indicating that the chemical reactions of the edible oil molecules are more intense and generate more polarized substances under high-temperature conditions. At the same time, under the same heating conditions, different types of edible oil have different increments in dielectric constant, which is due to the different compositions of fatty acids in each oil. However, the change trends of the dielectric constant are the same. Since the change of the loss factor has no effect on the operating frequency of the sensor and the influence on the return loss can be neglected, the dielectric loss of edible oil is not measured in this study. The sensor has good sensitivity in the quality detection of edible oil.

The dielectric permittivity of edible oil is affected by a variety of factors, among which the high temperature has a significant influence on it. When the edible oil is heated at high temperatures, its dielectric permittivity will increase while its quality will deteriorate. Figure 9a,b show the simulation results of S11 parameters of canola oil and soybean oil before and after heating, respectively. The operating frequency of the heated edible oil shifted about 10 MHz to the low-frequency direction than before heating.

## 4. Sensor Testing and Analysis

The sensor designed in this paper is shown in Figure 10, and its size is 70 mm × 70 mm × 1.4 mm. The holes located at the four top corners of the sensor are used to connect and secure the antenna patch at the top, the microchannel in the middle, and the substrate at the bottom. The reflection coefficient (S11) of the sensor measured by the vector network analyzer is shown in Figure 11. The comparison of the measured and simulated results of the S11 parameters is shown in Figure 12. While the measured resonance frequency of the sensor is 2.72 GHz and the return loss is −17.7 dB, the simulated resonant frequency is 2.68 GHz and the return loss is −10.4 dB. Compared to the simulated results, the measured resonant frequency shifts about 40 MHz to the low-frequency direction. These variations are due to errors introduced during the production of the sensor and parasitic capacitance added by the SMA connector. The measured return loss is 7.3 dB which is smaller than the simulated one. The amplitude of the measured return loss is greater than that of the simulated result because the model uses coarse frequency resolution to shorten the calculation time during the simulation. The performance of the sensor is unaffected by deviations from the intended frequencies and return loss since the change in the dielectric constant that the sensor measures is dependent on the shift of the resonant frequency. The voltage standing wave ratio (VSWR) of the sensor at the center frequency point is 1.1369. When VSWR < 2, the bandwidth of the resonant frequency is 2.6775 to 2.6813 GHz with a fractional bandwidth of 0.63%. Here, the no-load Q-factor of the proposed sensor is calculated by the following equation [25]
(3)$$Q=\frac{f_{0}}{f_{3\mathrm{dB}}}$$
where *f*_0_ is the resonant frequency; *f*_3dB_ is the 3 dB bandwidth measured from S11; and the calculated Q-factor is 37.36.

Figure 13a,b show the measured characteristics of the two edible oils before and after heating. It can be seen that the central frequency of the heated edible oil will shift towards a lower frequency than that of the same edible oil without heating. This is due to the dielectric permittivity of the edible oil will change after heating, so that its resonant frequency changes. The resonant frequency and reflection coefficient of the sensor are determined by the real and imaginary parts of the dielectric permittivity, respectively. Edible oils contain base oils and additives that can lead to the degradation of edible oil consumption after prolonged heating. Therefore, the imaginary part of the permittivity is not accurately accessible. When the composite permittivity designed in the microfluidic channel is different, the electric field is disturbed and the resonant frequencies shifted to a different position. During the measurement, the response of the sensor is monitored in real-time by a vector network analyzer that scans the microwave frequency range of interest. A stop-flow technique is adopted, where the liquid sample is injected into the microfluidic channel by a peristaltic pump, and then the flow is stopped for the measurement. When the microfluidic channel is filled with liquid, the capacitance of the sensor increases with the disturbance of the electric field. The increase in capacitance is caused by the increase in the complex permittivity of the liquid under test and eventually makes the sensor’s center frequency shift toward the lower frequency. The microwave measurement used in this design will not change the performance of the liquid under test, thus achieving nondestructive detection. Any change in the dielectric constant of the liquid causes a change in the transmission resonance. When testing the quality of edible oil, the room temperature is kept at about 25 °C. The microfluidic channel is washed and dried after measuring each liquid to avoid the influence of other factors on the measurement results. Compared with the results simulated by HFSS shown in Figure 6, it can be seen that the test results are more consistent with the simulation results, but there are still some differences. However, the resonant frequency will change with the different dielectric permittivity of the microchannel, which is consistent with the detection principle of the sensor. Therefore, it does not affect the performance of the sensor.

A microwave sensor’s sensitivity is defined as the ratio of the measured frequency offset (in %) in the reflected notch of an uncovered sensor to the offset of the real part of the complex permittivity [26], which is written as
(4)$$S=\frac{f_{\varepsilon_{r2}}-f_{\varepsilon_{r1}}}{(\varepsilon_{r2}-\varepsilon_{r1})f_{0}}\times 100\%$$
where *S* is the sensitivity of the sensor; $f_{\varepsilon_{r2}}$ and $f_{\varepsilon_{r1}}$ are the resonant frequencies corresponding to the sensor when the measured liquid dielectric constant is $\varepsilon_{r2}$ and $\varepsilon_{r1}$, respectively; and $f_{0}$ is the resonant frequency of the sensor when the microfluidic channel is unloaded.

From Formula (4), the maximum sensitivity of the sensor proposed in this paper can be obtained as high as 1.87%. Table 2 shows the sensitivity of this sensor and other sensors for measuring liquids. It can be seen that the proposed sensor in this paper has high sensitivity.

Table 3 shows the comparison between the measured and simulated results of the sensor’s operating frequency and return loss. During the measurements, each liquid under test was measured five times, and the differences between each measurement result were small. The average of the five results was finally used as the final measurement result to ensure the accuracy of the experimental results. The data in the table show that the measured operating frequency of the sensor is shifted by 40 to 45 MHz compared to the simulated results, while the measured results of return loss are 7.2 to 7.8 dB, smaller than the simulated results. There are three reasons for these differences. One is the error introduced in the sensor during the manufacturing process, another is the parasitic capacitance in the SMA connector, and the third is the influence of small fluctuations in the field temperature on the results during measurement. The offset of the center frequency of the sensor is 9 MHz before and after heating the simulated canola oil and soybean oil. While, the measured result shows that the offset of the center frequency of the sensor is 9 MHz and 10 MHz, respectively, which is more consistent with the simulation results. Because the sensor designed in this paper is mainly by judging the offset of the center frequency to determine whether the quality of the measured liquid has changed, although the measured results have a slight deviation from the simulated results, it does not affect the performance of the sensor.

The flexible PDMS microchannel used in this paper can not only enhance the sensitivity of the sensor but also can reduce the error caused by the air gap, to minimize the measurement error. In addition, the sensor can be reused and is easy to operate. If the microfluidic channel of the sensor is damaged, it can be replaced. The sensor is not only sensitive but also economical and practical.

## 5. Conclusions

A microfluidic microwave sensor loaded with a star-slotted patch for edible oil quality inspection is designed in this paper based on a microstrip patch antenna. Since the dielectric constant of edible oil changes after heating, the sensor can sensitively detect whether the edible oil was heated according to the change of the dielectric constant of the edible oil in the microfluidic channel. The resonant offset of the sensor is approximately 10 MHz, and the maximum sensitivity is 1.87%. The microfluidic channel can be removed and replaced, and the sensor only needs to use trace liquid to achieve sensitive detection. The sensor has the characteristics of simple manufacture, economic benefit, high sensitivity, no pollution, and so on. Therefore, it has a good application prospect in the quality monitoring of edible oil.

## Figures and Tables

**Figure 1 sensors-22-06410-f001:**
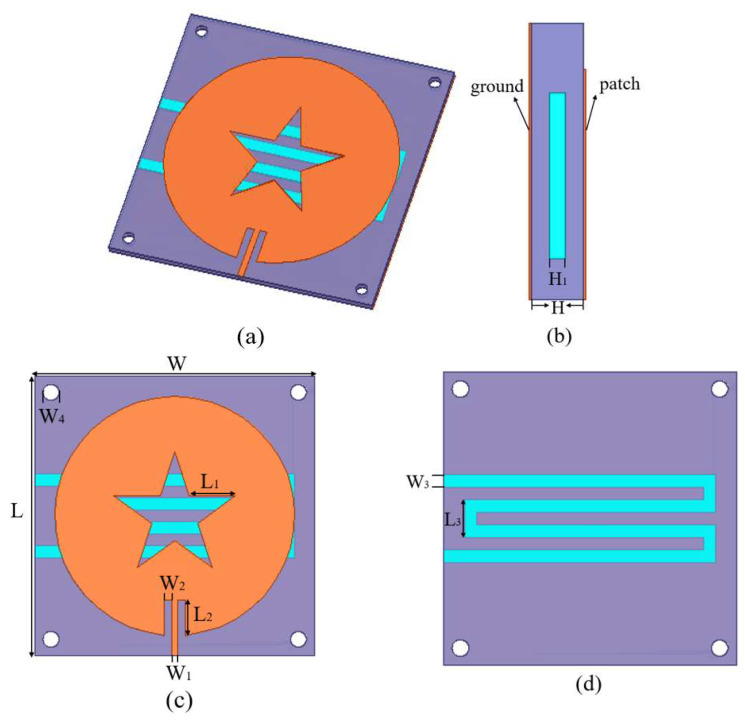
The structure of the microwave sensor: (**a**) 3D view; (**b**) side view; (**c**) the radiation patch; (**d**) the microfluidic channels.

**Figure 2 sensors-22-06410-f002:**
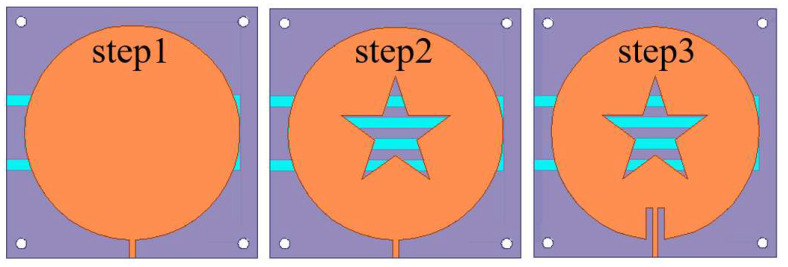
The evolution of the sensor patch slotting shape.

**Figure 3 sensors-22-06410-f003:**
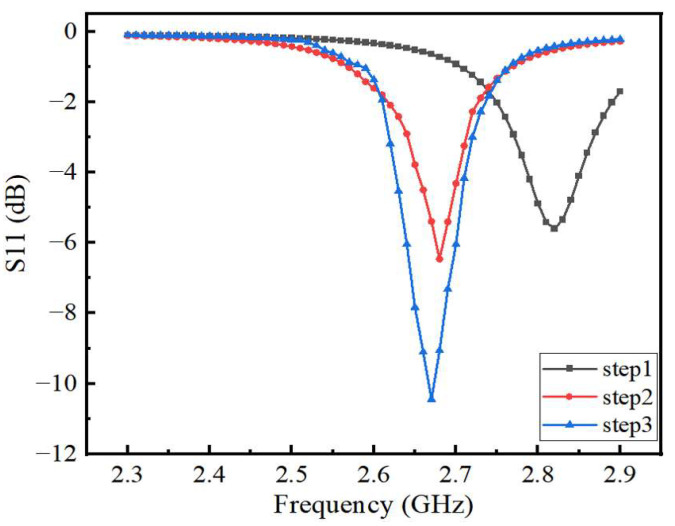
Comparison of the effect of slotted shape on the reflection coefficient of the sensor.

**Figure 4 sensors-22-06410-f004:**
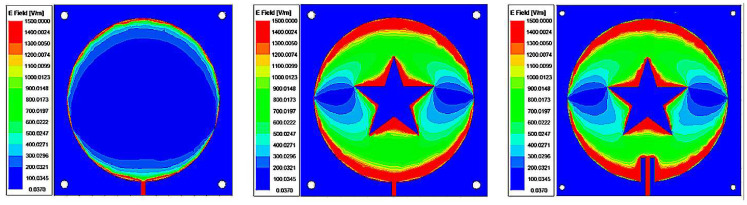
Electric field distribution at resonant frequency for different shaped patches.

**Figure 5 sensors-22-06410-f005:**
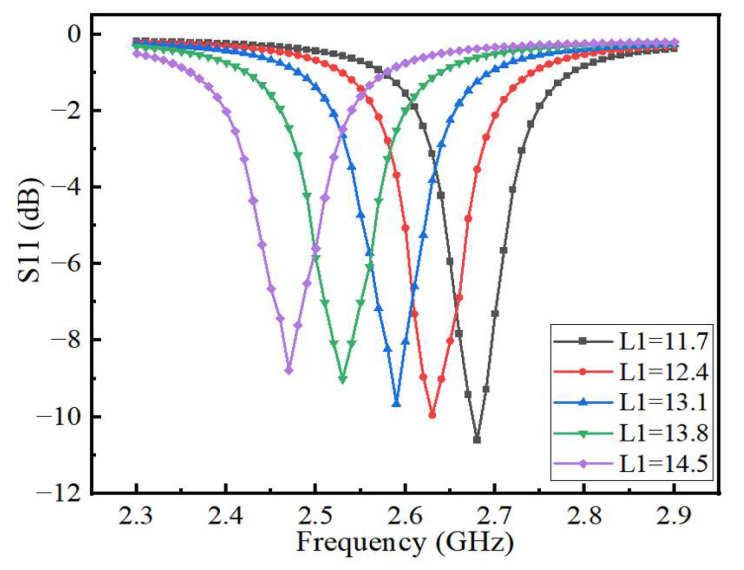
The effect of L1 on the resonant frequency at 2.68 GHz.

**Figure 6 sensors-22-06410-f006:**
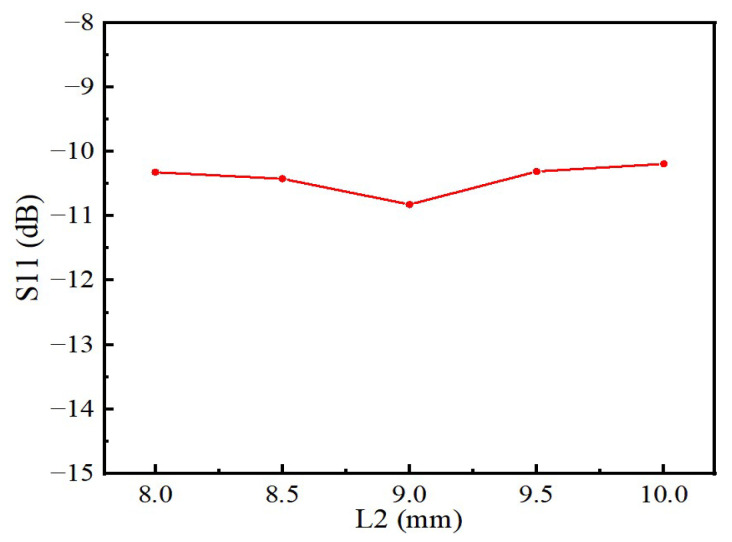
Influence of L2 on the reflection coefficient.

**Figure 7 sensors-22-06410-f007:**
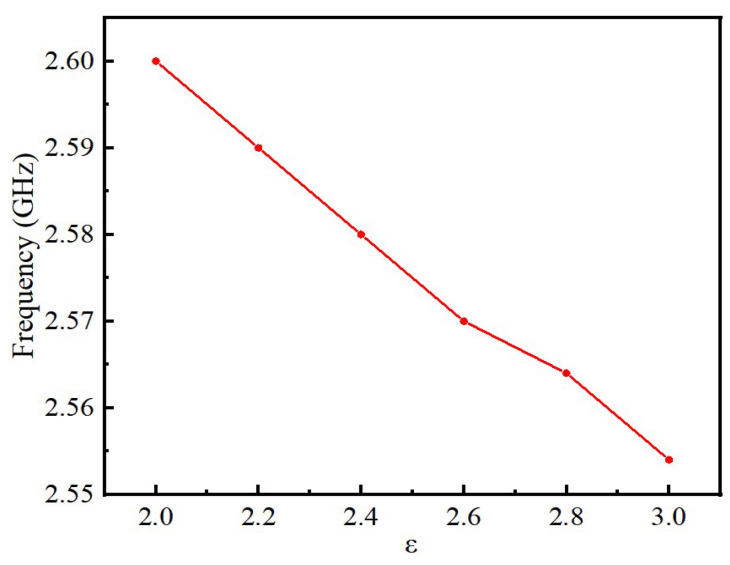
The relationship between the center frequency of sensor and the dielectric constant of microchannel.

**Figure 8 sensors-22-06410-f008:**
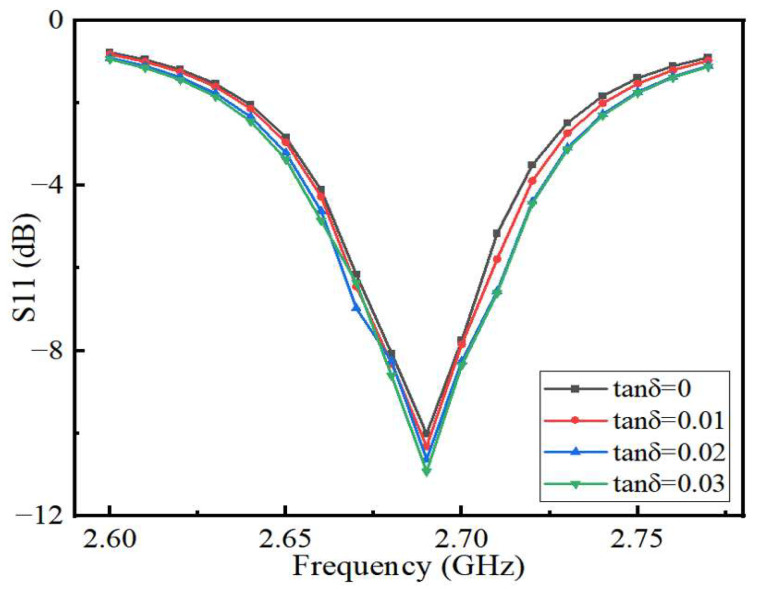
The relationship between S11 parameters of sensor and the variation of dielectric loss tangent angle of microchannel.

**Figure 9 sensors-22-06410-f009:**
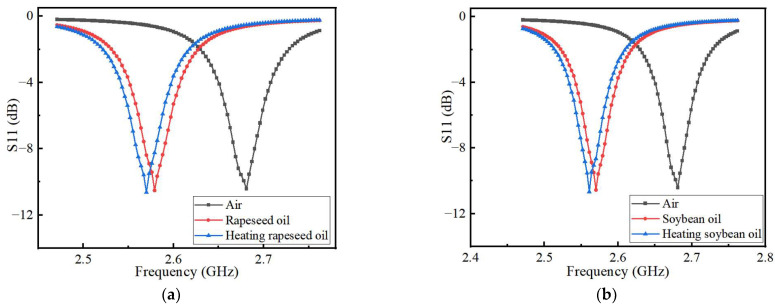
Simulation results of the effect of different (**a**) canola oil and (**b**) soybean oil cooking oil on the resonant frequency of the sensor before and after heating.

**Figure 10 sensors-22-06410-f010:**
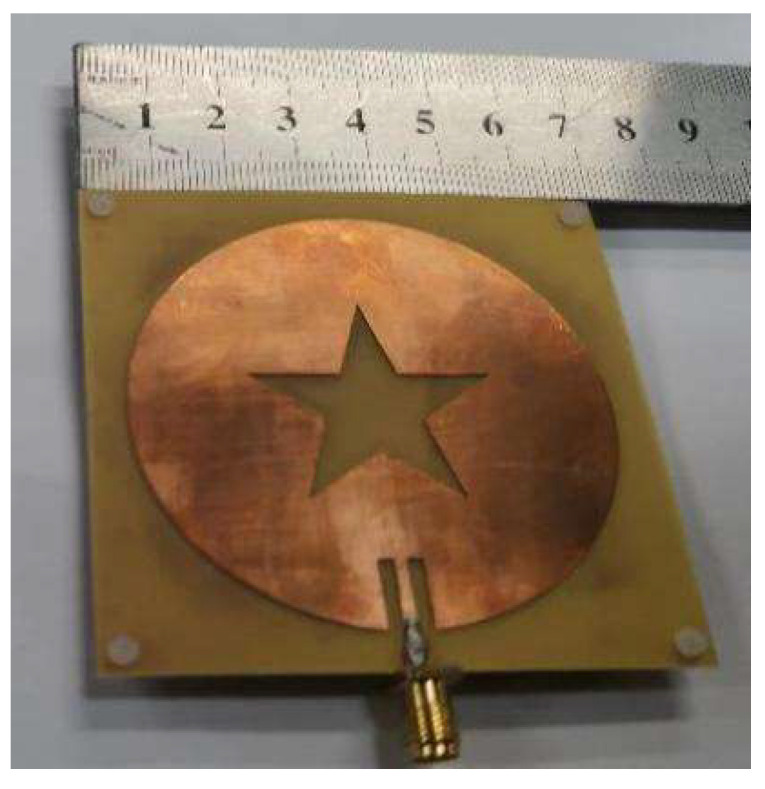
The fabricated Sensor.

**Figure 11 sensors-22-06410-f011:**
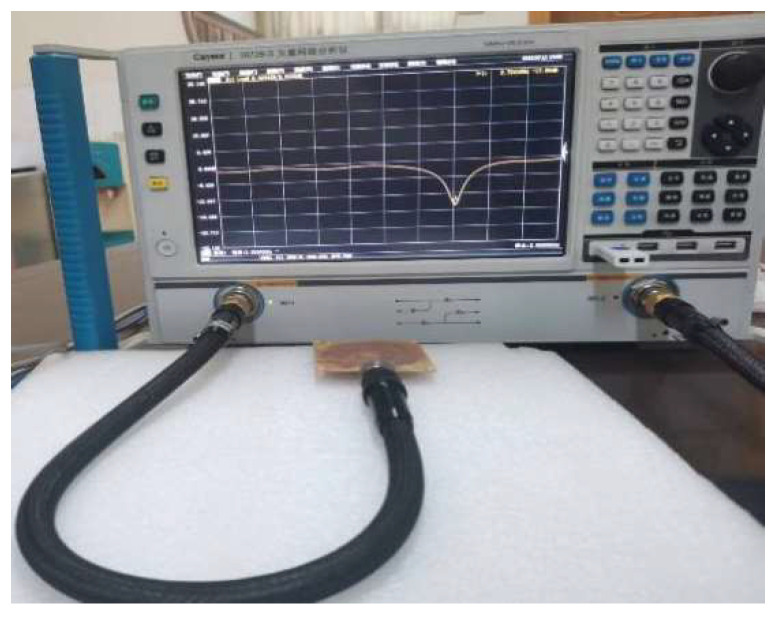
Diagram of measurement result.

**Figure 12 sensors-22-06410-f012:**
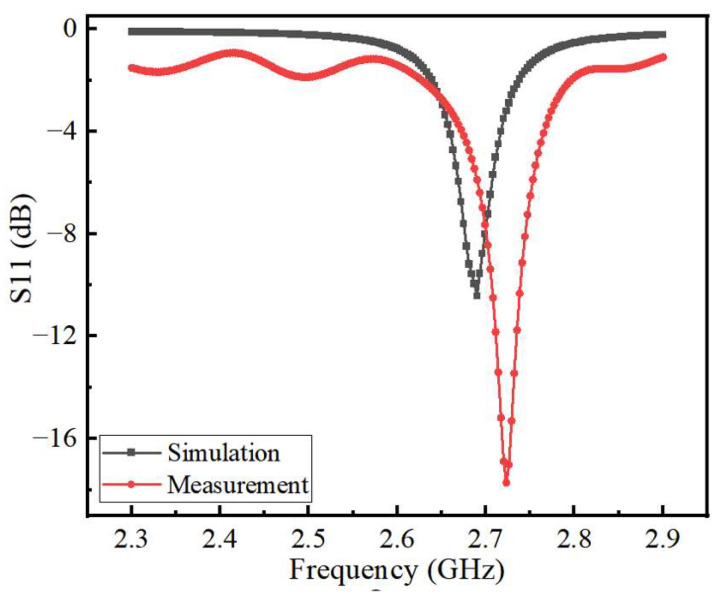
Plot of simulated versus measured results of S11 parameters.

**Figure 13 sensors-22-06410-f013:**
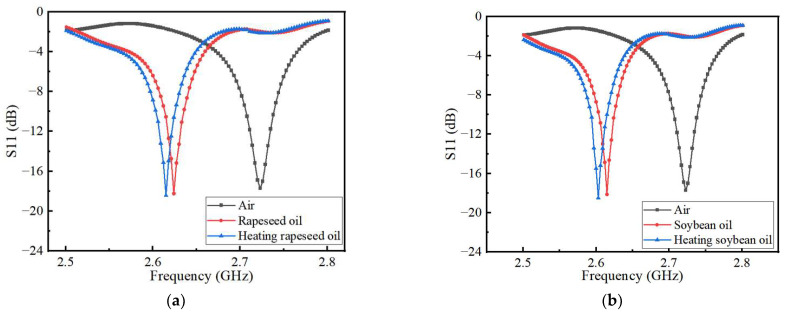
Measurement results of the effect of different edible oil on the resonant frequency of the sensor before and after heating (**a**) canola oil and (**b**) soybean oil.

**Table 1 sensors-22-06410-t001:** Dimensions of the sensor.

Parameter	Unit (mm)	Parameter	Unit (mm)
W L H L_1_ L_2_ L_3_	70 70 1.4 11.7 10 11	W_1_ W_2_ W_3_ W_4_ H	1.5 1.8 3 2 0.4

**Table 2 sensors-22-06410-t002:** Comparison of the sensitivity of this sensor with other sensors for measuring liquids.

Reference	*f*_0_ (GHz)	Sensor Structure	Sensitivity (%)
[26]	2.6	Metamaterial	0.27
[27]	0.87	SRRs	0.91
[28]	2.4	ring resonator	0.5
[29]	2.45	SRR	0.17
[30]	1.95	Metamaterial	0.2
[31]	3	SIRs	1.81
This work	2.68	round slot	1.87

**Table 3 sensors-22-06410-t003:** Comparison of sensor simulation and measurement results.

Tested Sample	Simulation		Measurement	
	Frequency (GHz)	Return Loss (dB)	Frequency (GHz)	Return Loss (dB)
Air	2.68	10.4	2.72	17.7
Rapeseed oil	2.579	10.53	2.624	18.2
Heating rapeseed oil	2.57	10.64	2.615	18.44
Soybean oil	2.571	10.56	2.615	18.14
Heating soybean oil	2.562	10.68	2.605	18.51

## Data Availability

Not applicable.

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
