# Peer review of "Microfluidic Microwave Sensor Loaded with Star-Slotted Patch for Edible Oil Quality Inspection"

_sensors, 2022, doi:10.3390/s22176410_

Round 1

Reviewer 1 Report

There is no justification on the shape of star. Authors should provide guide to the readers on why this shape is the optimum for sensing oil.  Is the star shape specific for oil? Does the shape affects sensitivity? Authors should explore this on simulations.

Authors refer to Figure 9 and Figure 10, but seem that these figures were not included in the manuscript, therefore the results are difficult to read and understand.  Authors should include all relevant figures.

Fig 8 Vs Fig 12a. Authors should explain why are they different (simulation vs experiment).

Authors make the following claim in the conclusion, however such claim is not supported by the results presented or the methods tested. "The microfluidic channel can be removed and replaced, and the sensor only needs to use trace liquid to achieve sensitive detection. "

Reviewer 2 Report

1.      The operating frequency range of designed sensor is with 2.3 to 2.9 GHz. The range is very narrow and might not able to investigate the performance of sensor at low and high frequency range. On the other hands, which result justify this? Figure 3 and 4 exhibit that only a particular design or dimension can be operated in single frequency. It does not exhibit a compatible frequency range from 2.3 to 2.9 GHz. If these frequencies refer to center frequency, what about the start and stop frequency that depict the range of frequency?

2.      Why the star shape is considered? Any advantages of this shape over the other? Please clarify.

3.      Please describe in detail about the mechanism of electromagnetic interaction with the sample under test. It is important in description of the working principle of sensor in sample detection.

4.      Fig. 6 has not been discussed in text. It seems it is ignorable.

5.      Why the center frequency declines when dielectric constant increases as shown in figure 7?

6.      Result of Fig. 7 can be verified via dielectric measurement using vector network analyzer.

7.      Resonant frequency (minimum peak) change for both oils should compare with air. Then, the Q- factor can be considered in your work.

8.      When the edible oil is heated at high temperatures, its dielectric permittivity will increase while its quality will deteriorate.”

Please explain the reason of the heated edible oil that led to increment of dielectric permittivity.

9.      The dielectric permittivity refers to dielectric constant? What about loss factor? Loss factor cannot be ignored within microwave spectrum.

10.   Please discuss the factors that lead to the difference between simulated and measured S11.

11.   Edible oils contain base oils and additives that can lead to the degradation of edible oil consumption after prolonged heating. Therefore, the imaginary part of the permittivity is not accurately accessible.”

The quoted reason is irrelevant to justify the exclusion loss factor from this work.

12.   Apart from the performance and specification of the sensor, the mechanism of interaction between applied field and sample is the paramount importance.

13.   Many other findings of results that need to be summarized in conclusion to accentuate the attainment of objectives.

Round 2

Reviewer 1 Report

Authors have addressed most of the comments accurately and sufficiently. The paper can be accepted without any further changes.